



# Ocean alkalinity enhancement using sodium carbonate salts does not impact Fe dynamics in a mesocosm experiment

David González-Santana[1], María Segovia[2], Melchor González-Dávila[1], Librada Ramírez[2], Aridane G. González[1], Leonardo J. Pozzo[2], Veronica Arnone[1], Víctor Vázquez[2], Ulf Riebesell[3], J. Magdalena Santana-Casiano[1].

[1]Instituto de Oceanografía y Cambio Global, IOCAG, Universidad de Las Palmas de Gran Canaria, ULPGC, Spain.
[2]Department of Ecology, Faculty of Sciences, University of Malaga, Málaga, Spain.
[3]GEOMAR Helmholtz Centre for Ocean Research Kiel, Kiel, Germany.

*Correspondence to*: David González-Santana (David.gonzalez@fpct.ulpgc.es)

**Abstract.** The addition of carbonate minerals to seawater through an artificial Ocean Alkalinization Enhancement (OAE) process increases the concentrations of hydroxide, bicarbonate, and carbonate ions. This leads to changes in the pH and the buffering capacity of the seawater. Consequently, OAE could have relevant effects on marine organisms and in the speciation and concentration of trace metals that are essential for their physiology. During September and October 2021, a mesocosm experiment was carried out in the coastal waters of Gran Canaria (Spain), consisting of different levels of total alkalinity (TA). Different concentrations of carbonate salts ($NaHCO_3$ and $Na_2CO_3$) previously homogenized were added to each mesocosm to achieve an alkalinity gradient between $\Delta 0$ to 2400 µmol L$^{-1}$. The lowest point of the gradient was 2400 µmol kg$^{-1}$, being the natural alkalinity of the medium, and the highest point was 4800 µmol kg$^{-1}$. Iron (Fe) speciation was monitored during this experiment to analyse whether total dissolved iron (TdFe), dissolved iron (dFe), soluble iron (sFe), dissolved labile iron (dFe´), iron-binding ligands (LFe) and their conditional stability constants ($K'_{FeL}$), could change because of OAE and the experimental conditions in each mesocosm. Observed iron concentrations were within the expected range for coastal waters, with no significant increases due to OAE. However, there were variations in Fe size fractionation during the experiment. This could potentially be due to chemical changes caused by OAE, but such effect being masked by the stronger biological interactions. In terms of size fractionation, sFe was below 1 nmol L$^{-1}$, dFe concentrations were within 0.5-4.0 nmol L$^{-1}$, and TdFe within 1.5-7.5 nmol L$^{-1}$. Our results show that over 99% of Fe was complexed, mainly by $L_1$ and $L_2$ ligands with $k'_{Fe'L}$ ranging between 10.92±0.11 and 12.68±0.32, with LFe ranging from 1.51±0.18 to 12.3±1.8 nmol L$^{-1}$. Our data on iron size fractionation, concentration, and iron-binding ligands substantiate that the introduction of sodium salts in this mesocosm experiment did not modify iron dynamics. As a consequence, phytoplankton remained unaffected by alterations in this crucial element.





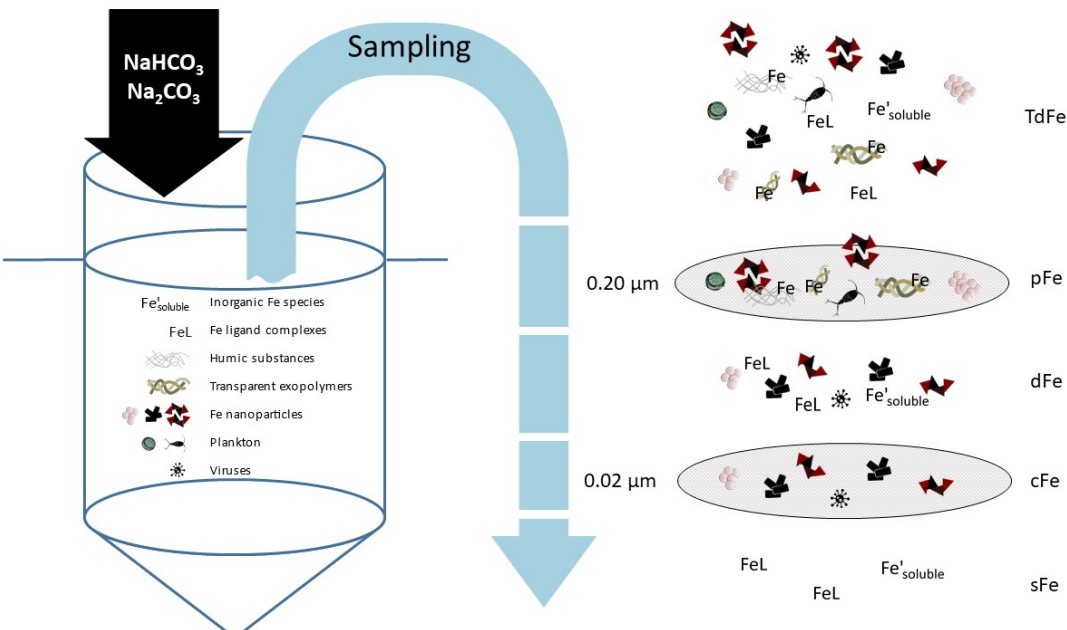


## 1. Introduction

Artificial ocean alkalinization enhancement (OAE) is a negative emission technology (NET) that can both produce the atmospheric carbon dioxide removal (CDR) by increasing the ocean carbon uptake and reversing ocean acidification (OA), favoring a higher ocean pH buffering capacity (Kheshgi, 1995; Renforth and Henderson, 2017; Lenton et al., 2018). OAE

can be likened to an acceleration of natural chemical weathering processes of minerals. Weathering reactions have played a role in modulating climate on geological time scales (Zeebe, 2012; Colbourn et al., 2015).

OAE might have consequences in the $CO_2$-carbonate system in the ocean. The increase in ocean alkalinity raises both total dissolved inorganic carbon (DIC) and pH, subsequently heightening the carbonate ion concentration, as well as decreasing $pCO_2$ and hydrogen ion concentration. This increases the $pCO_2$ imbalance between the ocean and atmosphere leading to an

enhancement of ocean carbon uptake. Moreover, the hoisted pH and carbonate ion concentration reverses ocean acidification due to the uptake of anthropogenic $CO_2$ from the atmosphere (Lenton et al., 2018). The artificial augmentation of alkalinity could also have consequences for marine ecosystems, both for the biota and for the chemical constituents of seawater, including elements that are essential for the ecosystem (Feng et al., 2016). The consumption of atmospheric $CO_2$ and pH increase during OAE may be accompanied by changes in the release of mineral dissolution products such as Si, Ca, Mg, Fe

or Ni. The ecological and biogeochemical consequences of OAE will also depend on the minerals used during alkalinization ($Mg_2SiO_4$, CaO, $Na_2CO_3$) (Renforth and Henderson, 2017; Hartmann et al., 2013; Bach et al., 2019). Hence, it is necessitated to perform experiments that may support or dissuade the use and viability of this technique at different experimental



conditions. One controlled, i.e. semi-natural, way to perform these types of experiments is by using mesocosms. Mesocosm studies allow rigorous testing of NETs at the ecosystem level since the only variable that is artificially modified is the factor
(OAE), allowing the enclosed water column to maintain its chemical and biological properties.

OAE could have an impact on trace metals that are essential for marine organisms and this effect has been scarcely investigated so far (Guo et al., 2022; Xin et al., 2023). Among metals, Fe is the most essential micronutrient controlling phytoplankton growth mainly through nitrate assimilation and photosynthesis (Behrenfeld and Milligan, 2013). Thus, the growth and development of marine phytoplankton is linked to the biogeochemical cycling of Fe (Jickells et al., 2005; Moore
et al., 2001), by regulating both the structure and the productivity of marine ecosystems (Tagliabue et al., 2017; Boyd and Ellwood, 2010).

The assimilation of iron by marine organisms is not straightforward. Fe bioavailability, i.e. iron available for uptake, and therefore growth, is controlled by many factors. It depends on the different forms (chemical speciation) in which iron is found in solution (Sutak et al., 2020). Indeed, not all iron species are bioavailable, only those in the dissolved (dFe) phase.
Under oxygenated ocean conditions, the most abundant form of iron is the insoluble form of oxidized iron, $Fe(III)$. pH is the main variable that controls Fe speciation in this condition. $Fe(II)$ is thermodynamically unstable and oxidized on timescales of minutes to hours (Santana-Casiano et al., 2005; González-Santana et al., 2021). Higher pH alters iron speciation, decreases Fe solubility and $Fe(II)$ half-life time, and thus decreases dissolved iron (dFe) concentrations (Liu and Millero, 2002; Boye et al., 2006; González-Dávila et al., 2006). However, there are several factors that can significantly alter and
control iron speciation, such as the photooxidation in surface waters, the presence of organic ligands (L) and the decreasing in the oxygen concentration (Santana-Casiano et al., 2022; González et al., 2019; Benner, 2011; Barbeau, 2006; Moffett, 2021; Hopkinson and Barbeau, 2007). All of them have in common that they can induce a reduction of $Fe(III)$ to $Fe(II)$ depending on the conditions and characteristics of the organic compounds involved. Moreover, organic matter plays an essential role, because it can complex up to 99% of the iron (Fe(III)-L) forming part of the dFe (Wu and Luther III, 1994;
Arnone et al., 2022). The concentration and strength of iron-binding organic ligands ($L_{Fe}$) modulate Fe speciation and its bioavailability. In addition, particle size (physical speciation) is key for Fe cycling, since the colloidal size remains in solution while larger fractions tend to sink down, being excluded from the system. All these processes were demonstrated during a mesocosm experiment, in a Norwegian fjord, that studied the effects of ocean acidification. In this experiment, both low pH and strong organic ligands enhanced the solubility of particulate and colloidal Fe (Segovia et al., 2017; Lorenzo et
al., 2020). This promoted a significant higher Fe availability to phytoplankton, directly impacting on their physiology and controlling the phytoplankton community structure in coastal ecosystems (Segovia et al., 2017; Lorenzo et al., 2020; Mausz et al., 2020).

The introduction of carbonate minerals into seawater by OAE (Bach et al., 2019) is expected to increase the concentrations of hydroxide, bicarbonate and carbonate ($OH^-$, $HCO_3^-$ and $CO_3^{2-}$). This will change metal ion speciation in seawater (Millero
et al., 2009). Metals that form complexes with $OH^-$ and $CO_3^{2-}$, such as Fe, may be less abundant in their free forms at higher pH. Moreover, $Fe(II)$ oxidation rate constants are strongly increased under high $HCO_3^-$ and $CO_3^{2-}$ concentrations (Santana-



Casiano et al., 2005, 2006) making Fe less bioavailable. AOE can also cause a significant release of $Ca^{2+}$ or $Mg^{2+}$ ions (Bach et al., 2019). These ions compete with Fe for specific functional groups of the organic ligand, which would affect the redox speciation of this metal (Santana-Casiano et al., 2010).

In this sense, the impact of OAE on the iron cycle could be that the Fe speciation will change with enhanced alkalinity. In the presence of organic compounds produced by the organisms in the mesocosms, the formation of strong organic complexes (FeL) will dominate the speciation of Fe. The $Ca^{2+}$ or $Mg^{2+}$ ions produced in AOE if $Mg_2SiO_4$, CaO were used, could compete with Fe for the organic ligands, which would affect the Fe redox speciation and the bioavailability. To avoid this effect, alkalinization with $NaHCO_3$ and $Na_2CO_3$ is preceptive.

The aims of this research were: 1) to study the evolution of inorganic and organic Fe speciation in seawater under OAE scenarios; and 2) to assess Fe bioavailability changes after Fe speciation which could potentially affect or be affected by the phytoplankton community. To obtain a clear response to these objectives, some considerations must previously be taken into account: 1) did the used alkalinization methodology act as a significant iron source?; 2) which are the effects of sodium carbonate salts on the iron pool size fractionation during OAE?; and 3) what is the effect of OAE on Fe-binding ligands

within the mesocosms?

## 2.  Material and Methods

### 2.1.  Mesocosms experimental design and distribution

The mesocosm experiment was conducted in the port of Taliarte, Telde, Gran Canaria during a 33-day period from September to October 2021. Nine mesocosms (M1-M9) were deployed in the port of Taliarte containing a natural seawater

column of 8.3 m³ each at 5 m depth, with a sediment trap attached to the bottom of the bag (Riebesell et al., 2013; Taucher et al., 2017) (Fig. 1) .



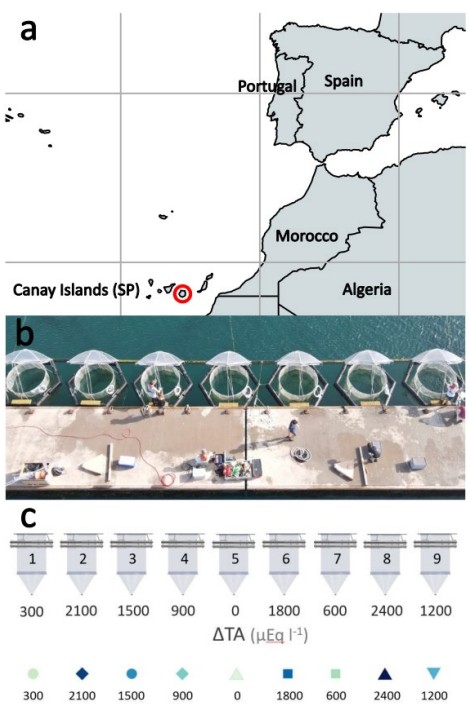

**Figure 1. Study site. a) Location in the Gran Canaria port town of Taliarte, Spain (© Google Erath Pro); b) Operating mesocosms during the experiment (photograph by Peter Yeung, National Geographic); c) Mesocosms experimental design representing the alkalinity gradient (OAE) (µmol L⁻¹) in the nine mesocosms: M5 (0), M1 (300), M7 (600), M4 (900), M9 (1200), M3 (1500), M6 (1800), M2 (2100), M8 (2400) (design by Silvan Goldenberg, GEOMAR).**

Mesocosms were set-up on September 10th, 2021 (day t0). Phase 0 (t1-t3) was established at the beginning of the experiment, prior to the addition of the sodium carbonate salts ($NaHCO_3$ and $Na_2CO_3$), to analyze the conditions of the mesocosms before their alkalinization. On day 4 (t4) of the experiment, 40 L of natural seawater with different concentrations of carbonate salts ($NaHCO_3$ and $Na_2CO_3$) previously homogenized were added to each mesocosm to achieve an alkalinity gradient between Δ0 (lowest) -2400 (highest) µmol L⁻¹. Phase I (t5-t19) began followed by Phase II (t19-t33). Variables were sampled at 1-day intervals. Experimental design representing the alkalinity gradient (OAE) (µmol L⁻¹) in the nine mesocosms was: M5 (Δ0), M1 (Δ300), M7 (Δ600), M4 (Δ900), M9 (Δ1200), M3 (Δ1500), M6 (Δ1800), M2 (Δ2100), M8 (Δ2400) µmol L⁻¹.

## 2.2. Sampling strategy

Samples were collected in 1 L acid-cleaned LDPE bottles. Trace metal clean (TMC) samples were subsampled from this bottle within an ISO Class-6 laminar flow hood (Cutter et al., 2017). Unfiltered samples (i.e., TdFe) were first collected. Subsequently, 0.2 µm filtered samples (Sartobran™ PES; i.e., sFe, dFe and LFe) followed on. Soluble Fe samples were further filtered through acid-cleaned 0.02 µm filters (Whatman™ Anotop™ 25/0.02) using a peristaltic pump. The TdFe,



dFe and sFe samples were acidified with HCl Ultrapure, 2 ‰ v/v to pH ~1.7 (Panreac) after sampling) and kept in the dark at least 6-months until analysis. The LFe samples were frozen to -20°C until further analysis was performed.

### 2.3. Iron concentrations analyses

The TdFe, dFe and sFe samples were analyzed in duplicates (two analytical peaks; a second duplicate was performed if the standard deviation was >5 %) using flow injection analysis with chemiluminescence detection (FIA) (Obata et al., 1993;
Lohan et al., 2006) inside an ISO Class-6 laminar flow hood inside an ISO Class-5 TMC laboratory. Samples were spiked with 0.013 M ultrapure $H_2O_2$ (Sigma-Aldrich) 30 min prior to analysis to ensure the complete oxidation of Fe (II) to Fe (III) (Lohan et al., 2006). Each sample was buffered in-line to pH 3.5 with 0.15 M ammonium acetate (Supelco and Sigma-Aldrich, SpA) before Fe(III) was preconcentrated onto the cation exchange resin Toyopearl-AF-Chelate 650 M (Tosohaas) between 60 and 120 s at a flow rate of 1.5 ml min$^{-1}$. Following a rinse step of weak 0.013 M HCl (Honeywell Fluka™, SpA),
Fe was eluted from the resin using 0.24 M HCl (Honeywell Fluka™, SpA) and entered the reaction stream where it was mixed with a 0.015 mM luminol solution containing 70 µl L$^{-1}$ triethylenetetramine (Sigma-Aldrich), buffered to pH 9.5±0.1 using a 1 M ammonia solution (Supelco, SpA). Iron concentrations were quantified using standard additions (TraceCERT) to low Fe seawater. The limit of detection (three times the standard deviation of the lowest addition) was 0.03±0.02 nmol L$^{-1}$ while the precision of three analytical peaks was <2 %. Accuracy was established by repeat quantification of dFe in an
inhouse standard, whose concentration was asserted from the repeat measurement of GSC reference samples yielding 1.59±0.03 nmol L$^{-1}$, which agrees with the reported consensus values (1.54±0.12 nmol L$^{-1}$).

### 2.4. Fe-binding ligands

Fe-binding ligands were measured by competitive ligand exchange-adsorptive cathodic stripping voltammetry (CLE-ACSV), following the method reported by (Croot and Johansson, 2000). Briefly, 10 mL samples were pipetted into TMC Teflon
bottles with 100 µL of EPPS (final concentration 10$^{-2}$ mol L$^{-1}$) and different concentrations of Fe were added (from 0 to 15 nmol L$^{-1}$). After a 1 h equilibration period, 10 µL of TAC (final concentration 2 µmol L$^{-1}$), the titration series were measured in a Teflon cell, with two +0 Fe additions and more than 10 titration points (Gledhill and Buck, 2012; Garnier et al., 2004). The dFe-binding ligands (L$_{Fe}$), and the conditional stability constants (K$_{condFeLi}$, meaning the strength of a metal ion-ligand interaction) were computed using the ProMCC software (Omanović et al., 2015).

Iron stock solutions were prepared weekly from standard solutions for atomic absorption spectrometry (Fluka), diluted with MQ-water and acidified with 100 µL ultrapure 12.8 M HCl. A 1M stock buffer of EPPS (N-(2-hydroxyethyl) piperazine-N′;2-propanesulfonic acid; *Sigma-Aldrich*) was prepared in 1M ultrapure NH$_4$OH (VWR) at pH 8.2. Possible remaining metals were removed from the buffer solution by adding 100 µM of MnO$_2$, stirred overnight and filtered through an acid-clean PES 0.45 µm filter (Campos, M. Lucia and van den Berg, 1994). As a competitive ligand, 0.01 M stock solution of
TAC (2-(2-thiazolylazo)-p-cresol; Sigma-Aldrich) was prepared in methanol (Sigma-Aldrich). All stock solutions were kept in the fridge at 8°C under darkness).



## 3. Results and discussion

The experiment was divided into three distinct phases according to characterized variations in the seawater chemistry and the plankton community development, as indicated above. A phase-0 was established at the beginning of the experiment (t1-t3) to analyze the conditions in the mesocosms prior to treatment additions (OAE). Phase-I (t5-t19) began after alkalinization by NaHCO$_3$ and Na$_2$CO$_3$ and then phase-II followed (t19-t33).

The experimental setup effectively recreated a scenario of ocean liming by establishing a gradient of total alkalinity (TA) increasing in 300 µmol L$^{-1}$ intervals. Dissolved inorganic carbon (DIC) and TA were steady until day t21 (Fig. S1-supplemental material). On that day, in the highest treatment (Δ2400 µmol L$^{-1}$), abiotic precipitation occurred but, this event did not have any impact on the phytoplankton community response. A succinct summary of the phytoplankton abundance results is provided next and in Fig. S2-supplemental material. During Phase-I, there was a significant increase in picoeukaryotes abundances at Δ600, Δ900, Δ1800, and Δ2100 µmol L$^{-1}$. Additionally, microplankton increased at Δ900 µmol L$^{-1}$. In contrast, small nanoeukaryotes (2-20µm) exhibited a decreasing trend from the experiment's onset, with maximal abundance observed before OAE treatment additions (t1-t3). Moving to Phase-II, large nanoeukaryotes (> 20µm) dominated the community, particularly in Δ1500 and Δ1800 µmol L$^{-1}$. Microplankton also increased by the phase's end. *Synechococcus spp.* showed a general increase in cell density over time, peaking at t27 in Δ900 µmol L$^{-1}$, but subsequently decreasing in many other treatments. All this data can be consulted for further details and in the articles by Marin-Samper et al. and Ramirez et al., this issue.

### 3.1. Iron size fractionation

Iron size fractionation (sFe, cFe, dFe and TdFe) was determined during the OAE experiment (Fig. 2). The first metal concentration samples were collected on t1 for dFe and sFe, i.e. phase-0 of the experiment. The initial concentration for dFe was 1.74±0.44 nmol L$^{-1}$, ranging from 1.20±0.02 nmol L$^{-1}$ at Δ1800 to 2.39±0.03 nmol L$^{-1}$ at Δ1500. For sFe, the initial concentrations were 0.39±0.18 nmol L$^{-1}$, ranging from 0.22±0.01 nmol L$^{-1}$ at Δ600 to 0.74±0.03 nmol L$^{-1}$ at Δ1800.



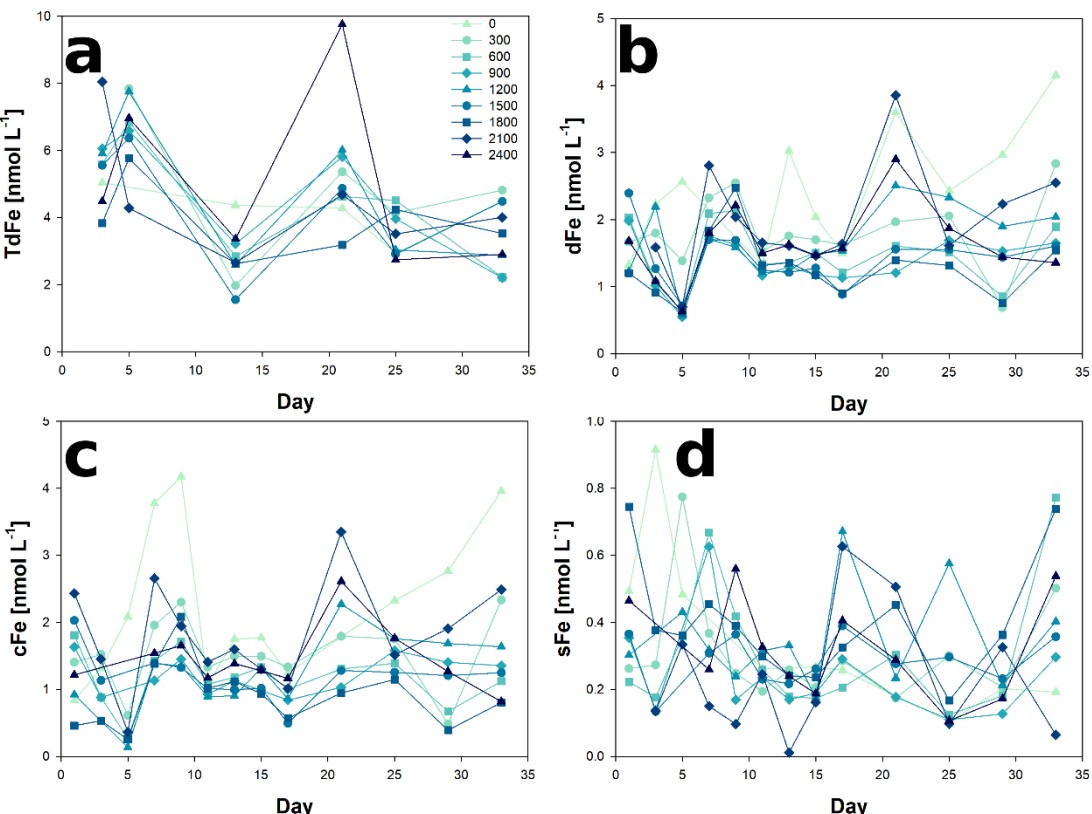

**Figure 2. Temporal evolution of a) total dissolvable iron (TdFe), b) dissolved iron (dFe), c) colloidal iron (cFe = dFe-sFe) and d) soluble iron (sFe) concentrations from day 0 to day 33 of the mesocosm experiment for the nine increased alkalinity mesocosms ranging from 0 (light blue) to 2400 µmol L$^{-1}$ (dark blue). Symbols and color-codes as in figure 1C. The concentration range of the different size fractions is different. Three y-axis ranges have been used; up to 10 nmol L$^{-1}$ for TdFe, up to 5 nM for dFe and cFe, and up to 1 nmol L$^{-1}$ for sFe.**

In the Macaronesia region, dFe concentrations average 0.80±0.25 nmol L$^{-1}$ (varying from 0.46 to 1.32 nmol L$^{-1}$) (Arnone et al., 2022). However, these concentrations are obtained from samples collected on a research vessel using trace metal clean conditions into a trace metal clean bottle following strict trace metal clean conditions (Cutter et al., 2017). As such, t1 dFe concentrations just show a slight increase in dFe when no trace metal clean procedures were followed and mesocosms were kept in the open air, but still trying to limit contamination. The observed changes in dFe and sFe concentrations were most likely due to the natural variability between mesocosms. Considering that variability occurs on interannual, seasonal and diurnal time scales, this also forms part of the system's behavior (Schulz and Riebesell, 2013). Indeed, considerable natural variability exists in seawater carbonate chemistry speciation, caused by changes in temperature and biological activities such as photosynthesis, respiration, nutrient utilization, remineralization and calcium carbonate precipitation and dissolution



(Segovia et al., 2017). These processes were highly active during our experiment, especially during phase-II (Paul et al;
Marin-Samper et al; this issue) thus probably directly affecting metal speciation.

The mesocosm with Δ2100 was not included in the calculated means due to an outlier in both dFe = 5.30±0.10 nmol L-1 and
sFe = 2.86±0.05 nmol L$^{-1}$. The first TdFe samples were taken on day t3 and averaged 6.26±2.4 nmol L$^{-1}$. The high standard
deviation in the results was caused by Δ300 and Δ2100. When these two mesocosms were excluded, the average
concentration decreased to 5.21±0.81 nmol L-1.

On day t4, sodium carbonate salts, NaHCO$_3$ and Na$_2$CO$_3$, were added to the mesocosms. and phase-II started. Mesocosms in
which ΔTA was greater than 300 μmol L$^{-1}$ showed decreases in dFe. The resulting dFe concentrations were 0.63±0.06 nmol
L$^{-1}$ for all the mesocosms except for ΔTA 0 and 300 μmol L$^{-1}$ (2.56±0.02 and 1.39±0.08 nmol L$^{-1}$, respectively). These
concentrations were the lowest observed throughout all the experiment and showed the least variation between mesocosms at
any timepoint. The decrease in dFe was most likely caused by a decrease in colloidal sized Fe (cFe). Aggregation processes
did not affect the sFe, which remained relatively unchanged. Mesocosms 5 and 1 (ΔTA 0 and 300 μmol L$^{-1}$ respectively)
showed higher concentrations but were within the analysis range observed during the mesocosm experiments.

Two days after the alkalinization event, dFe concentrations returned to the original concentration range (2.00±0.39 nmol L-
1). This change in concentration back to normal did not correlate with the addition of NaHCO$_3$ and Na$_2$CO$_3$ required for the
OAE. The increase in dFe was not accompanied by an increase in sFe along phase-I . It was the cFe fraction that varied. This
suggests that aggregation processes of colloidal particles occurred with other particles with limited adsorption/absorption of
sFe. TdFe also increased after the addition of NaHCO$_3$ and Na$_2$CO$_3$. This increase could be due to multiple reasons including
1) the opening of the mesocosms, which could have allowed for aerial input of iron, 2) the addition of Fe during the addition
of NaHCO$_3$ and Na$_2$CO$_3$ including possible trace amounts in the used reagents, 3) the lack of trace metal clean conditions, 4)
the presence of mesozooplankton faecal pellets, and 5) the aggregation of cFe and dFe could have led to the formation of
larger size particles (>0.2 mm) that could have been dissolved during the 6 months of sample acidification, compared to
highly refractory particles that may not solubilize in 2‰ HCl.

During phase-II (t$_{19}$ – t$_{33}$), Fe size fraction concentrations do not present any significant correlation (p>0.05) with biological
parameters measured during the OAE or any correlation with the ΔTA. Their variability

This mesocosm experiment was carried out in coastal waters where Fe is not a limiting micronutrient (López-García et al.,
2021; Arnone et al., 2022). However, Fe redox processes should be considered in future OAE where iron can become a
limiting element. For example, Fe(II) is considered the most bioavailable Fe species due to its less energetically costly
uptake by organisms. However, in the current oxic ocean, Fe(II) is rapidly oxidized to Fe(III) (Santana-Casiano et al., 2005).
The addition of NaHCO$_3$ and Na$_2$CO$_3$ reduced the cFe concentration. Recent studies of Fe(II) oxidation kinetics have shown
that colloidal size particles play a significant role in Fe(II) oxidation kinetics (González-Santana et al., 2021). Indeed, a
decrease in colloids drastically accelerates the Fe(II) oxidation rate which subsequently, could lead to a decrease in available
Fe(II), reducing Fe bioavailability and Fe acquisition costs in terms of energy (microbial/enzymatic catalysis) (Fujii et al.,
2010; Anderson and Morel, 1982).



Correlations between the different iron size fractions and their ratios, mesocosm physico-chemical parameters (i.e., pH, alkalinity, dissolved inorganic carbon, nitrates, nitrites, silicates) and biological variables (i.e., fluorescence, community respiration, net primary production, pico and nanoeukaryotes concentration) did not present any statistical significance (Fig. S3). Initial statistical tests showed a significant inverse correlation ($p < 0.05$ and $r > -0.5$) between TdFe and DIC and TA, however, when further tests were performed for phase-II, correlation decreased ($p > 0.28$ and $r = -0.15$). These results show that iron concentrations were not limiting during the experiments and so any of the phytoplankton functional groups were below their Fe demand for growth (Segovia et al. 2017). This is a direct result of high iron concentration as a consequence of the study site being located in coastal waters with high Fe inputs such as sediment resuspension in coastal waters and aerial inputs due to the Canary Islands location. Yet, such Fe concentrations were not toxic either, since phytoplankton growth was not hampered in phase-II. In fact, the contrary occurred, and Chlorophyll-a (Chla) levels increased in parallel with a significant recovery in the Fv/Fm ratio (Ramirez et al. this issue). These improvements in Chla and photophysiology data are primarily attributed to the emergence of blooms of large nanoeukaryotes and picoeukaryotes in $\Delta 1500$, $\Delta 1800$, $\Delta 600$ and $\Delta 900$ at t27.

In light of this, we suggest that iron size fractions and phytoplankton are independent. However, we have more than a reasonable doubt about the colloidal fraction being directly linked with mesozooplankton fecal pellets. For instance, >200% of a mixture of lithogenic and biogenic Fe seems to be egested via Antarctic mesozooplankton fecal pellets, while only 5% of the body iron content is assimilated (Schmidt et al., 2016). In any case, more work is needed to clarify this particular point.

### 3.2. Iron ligand distribution

Considering that the OAE were not carried out under Trace Metal Clean conditions, according with the GEOTRACES protocol (Cutter et al., 2017), except for both sample collection and measurement in the ULPGC facilities, the OAE experiments have demonstrated non-linearity in the behavior of Fe-binding ligands ($L_{Fe}$) and the conditional stability constant ($\log K^{cond}_{Fe'L}$) across the entire alkalinity gradient (Fig. 3). Fe-binding ligands in the ocean come from different sources such as rupture of cells after grazing (Sato et al., 2007), viral lysis (Poorvin et al., 2011), transformation of organic matter (Gerringa et al., 2006), phytoplankton exudates (Rico et al., 2013; Santana-Casiano et al., 2014) and sediments (Gerringa et al., 2008). The $L_{Fe}$ are classified according to the value of the conditional stability constants ($\log K^{cond}_{Fe'L}$). These ligands can be considered to be strong ($L_1$) or weak ($L_2$), where $\log K^{cond}_{Fe'L} = 12$ divides $L_1$ and $L_2$-type of Fe-binding ligands (Gledhill and Buck, 2012).





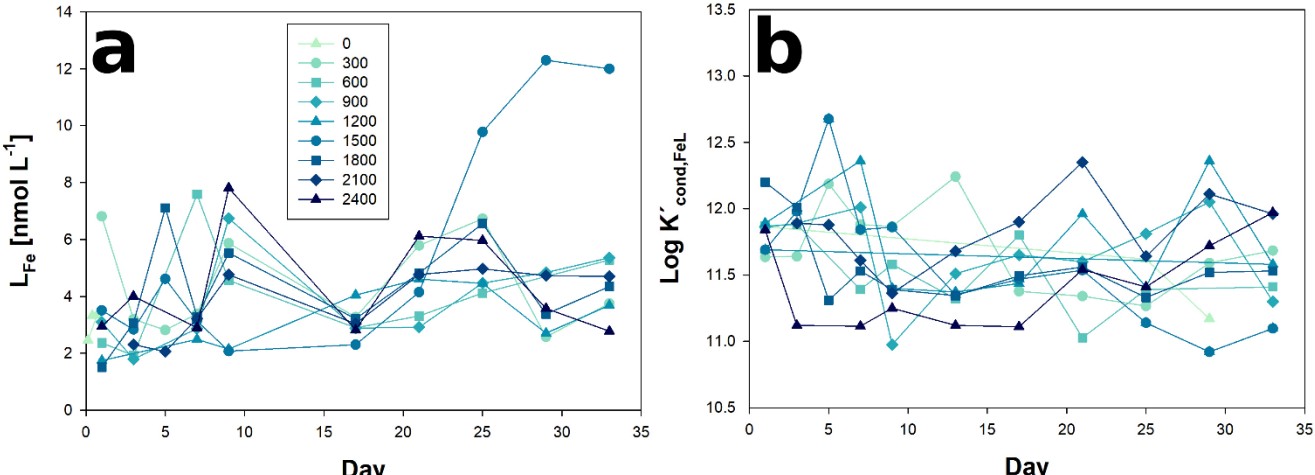

**Figure 3. Temporal evolution of a) iron ligands (LFe), and b) Conditional contant (K$_{cond}$), from day 0 to day 33 of the mesocosm experiment for the nine increased alkalinity mesocosms ranging from 0 (light blue) to 2400 μmol L$^{-1}$ (dark blue). Symbols and color-codes as in figure 1C.**

Specifically, previous to OAE (phase-0), L$_{Fe}$ concentrations ranged from 6.81 nmol L$^{-1}$ (Δ300) to 1.51 nmol L$^{-1}$ (Δ1800). Throughout the course of the experiments, the mean L$_{Fe}$ concentration across all mesocosms was 4.14±2.01 nmol L$^{-1}$, with a log K$^{cond}_{Fe'L}$ value of 11.63±0.36, being L$_2$-class in most of the cases (90.5% of the 84 data). Notably, there was a discernible alteration in the behavior of L$_{Fe}$ concerning Δ1500, coincident with the emergence of blooms of the Haptophyte *Chrysochromulina tobinni* at t27 (Marin-Samper et al., this issue). L$_{Fe}$ consistently exhibited higher concentrations than dFe,

with the exception of mesocosm Δ2100 at t1, Δ900 at t5, Δ600 at t5 and t27, Δ2400 at t5, and Δ1200 at t3 and t5. In these instances, the titration curves were linear, suggesting oversaturation of the natural ligands with the Fe content. Consequently, the percentage of Fe-organically complexed (%FeL) was determined to be 99.8±0.2. The concentration of labile Fe (Fe') was in the range from 0.04 to 15.98 pmol L$^{-1}$, and Fe$^{3+}$ species, primarily Fe(OH)$_3$, was computed within the concentration range of 1.6x10$^{-21}$ to 3.9x10$^{-26}$ mol L$^{-1}$.

It is important to remark that the experiment was not conducted under trace metal conditions. Consequently, discussing the potential Fe limitation in relation to the measured concentrations becomes a complex task. Additionally, the possibility of Fe contamination during the sampling process for other variables, which may influence Fe concentration and Fe-binding ligands, cannot be definitively ruled out.

The initial levels of dFe and L$_{Fe}$ at the start of the experiments were found to be comparable to those observed in coastal

waters previously documented by Arnone et al. (2022) in the Canary Islands. Specifically, the mean dFe concentration in the region was determined to be 0.80±0.16 nmol L$^{-1}$, with corresponding mean L$_{Fe}$ levels of 1.28±0.21 nmol L$^{-1}$. The initial L$_{Fe}$ concentrations within the OAE mesocosms ranged from 1.51 to 6.81 nmol L$^{-1}$, and their evolution over time fell within the



range of 1.51 to 12.30 nmol L-1. These values were also in line with data from surface waters of the Atlantic Ocean, ranging from 0.1 to 0.8 nmol L$^{-1}$ (Boye et al., 2006; Rijkenberg et al., 2008).

Measurement of $L_{Fe}$ within the mesocosms indicated a consistent nature of Fe-binding ligands, as reflected by the comparable values of the log $K^{cond}_{Fe'L}$ throughout the entire study period, within the range of 10.91 – 12.68, corresponding to the $L_2$ and $L_1$-class, being the weaker ligands the most abundant (90.5% of the samples). In the surrounding waters of the Canary Islands, log $K^{cond}_{Fe'L}$ values were reported to range between 10.77 and 11.90 (with a mean value of 11.45±0.29) (Arnone et al., 2022), aligning with surface ocean waters in the region (log $K^{cond}_{Fe'L}$ 8.8 – 12.85) (Boye et al., 2006; Bundy et

al., 2015; Rijkenberg et al., 2008; Thuróczy et al., 2010; Boye et al., 2003).

The ratio of $L_{Fe}$ to dFe serves as an indicator of the saturation state of ligands and whether conditions favor precipitation (Thuróczy et al., 2011). The observation of an excess of $L_{Fe}$ with values exceeding 1 nmol L$^{-1}$ may suggest a biological production of ligands. Thuróczy et al. (2011) previously reported a ratio of approximately 4.4 for surface waters characterized by high phytoplankton activity. In the case of our OAE experiment, the production of Fe-binding ligands may

be attributed to the increase in large nanoplankton and microplankton biomass as described by Ramírez et al. (this issue) during phase-II. Additionally, the presence of cyanobacteria, during the OAE can lead to the production of siderophores, as indicated by log $K^{cond}_{Fe'L}$ values falling between 11 and 14 (Bundy et al., 2018; Witter et al., 2000). Siderophores are organic molecules from microbial origin that bind metals with different affinities. They are thought to play an important role in iron cycling in the ocean, but relatively few marine siderophores have been identified. Recently, new siderophores produced by

*Synechococcus sp.* PCC 7002 have been discovered and characterized (Boiteau and Repeta, 2015) which is of relevance considering that *Synechococcus spp.* peaked during phase-II.

Furthermore, other types of ligands, such as polyphenolic compounds, may also be present, consistent with their log $K^{cond}_{Fe'L}$ values (Arreguin et al., 2021; González et al., 2019). Weak Fe-binding ligands (log $K^{cond}_{Fe'L}$ <12) could potentially be produced during the grazing of phytoplankton and the bacterial remineralization of sinking organic particles (Poorvin et al.,

2011; Sato et al., 2007).

The observed variability in labile Fe concentrations (Fe') within the range of 0.04 to 15.98 pmol L$^{-1}$ aligns with the findings of Arnone et al. (2022). Notably, the limited concentration of $Fe^{3+}$ precludes the favored precipitation of iron hydroxides. It has been established that the presence of organic binding ligands enhances the solubility of Fe, as demonstrated by previous research (Liu and Millero, 2002; Segovia et al., 2017).

Similar to iron size fraction concentrations, $L_{Fe}$ and $K^{cond}_{Fe'L}$ did not present any statistical correlation with any other measured physico-chemical and biological parameter. This presents a scenario where OAE in the working conditions does not affect the Fe concentration, fractionation, and speciation.



## 4. Conclusion

The OAE experiments, while not conducted under stringent trace metal conditions according to the GEOTRACES protocol, have provided valuable insights into the non-linear behavior of Fe-binding ligands ($L_{Fe}$) and the conditional stability constant (log $K^{cond}_{Fe'L}$) across the entire alkalinity gradient. $L_{Fe}$ concentrations increased 2-fold just after OAE with a log $K^{cond}_{Fe'L}$ throughout within the range of 10.91 – 12.68. This suggests the possible biological production of ligands, particularly during the emergence of nanoeukaryotes blooms and peaks of Synechococcus, as well as zooplankton grazing (Smith et al, this issue). Similarly, dFe concentrations were higher than those one might expect in the Macaronesia region. This could have been a result of the manipulation required to fill in mesocosms in the open environment. Where the iron sources such as sediment resuspension, aerial input and water runoffs exert a direct influence combined with our mesocosm manipulation. During the OAE we did not observe any iron size fractionation changes and concentrations variations due to different OAE conditions. Nevertheless, the addition of sodium carbonate salts ($NaHCO_3$ and $Na_2CO_3$) did produce decreases in the dFe concentration just after the addition. However, concentrations returned to background level concentrations in 48 h. Soluble iron remained below 0.8 nmol $L^{-1}$ through the whole experiment and did not present changes after the sodium carbonate salts addition. The results suggest that sodium carbonate salts addition will shortly affect cFe, but quickly return to original concentrations in coastal waters.

This OAE experiment was based on the addition of sodium carbonate salts. However, one might wonder which are the pros and cons of this method in comparison to other weathering-based CDRs. For instance, similar alkalinity concentrations could have been obtained by adding calcium or magnesium carbonate salts. However, a previous study showed that the addition of calcium or magnesium salts would have led to changes in the iron speciation (Santana-Casiano et al., 2010).Also open ocean dissolution of the iron-containing mineral olivine was simulated by using a marine carbon cycle model (Hauck et al. (2016). They found that after equilibration with the atmosphere, the addition of olivine did not enough compensate for the effect of ocean acidification and, in the real world experiments, olivine-prompted silicic acid and iron fertilization might have undesired effects since increased dFe does not always have beneficial consequences for the community (Coale, 1991; Behrenfeld and Milligan, 2013).

The iron size fractionation, concentration and iron-binding ligands data obtained supports the fact that the addition of sodium salts in this mesocosm experiment did not lead to significant changes in the iron cycle, i.e., did not alter the Fe cycle, therefore phytoplankton was not affected by changes in this essential element. This may have wider environmental implications in large-scale experiments and implementation of CDRs in larger ocean extensions. However, we must have in mind that whilst the use of this OAE technique in coastal waters could potentially not alter the iron cycle, the results could vary if the experiment was to be performed in oligotrophic open ocean waters, if a strong bloom, larger than what was observed in these experiments occurred, or if on the contrary, excess Fe might hold back phytoplankton growth due to toxic effects. Thus, more studies on the effects of CDRs on trace metals are needed to avoid unintended side effects on the ecosystem.



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

Data availability

The raw data supporting the conclusions of this article will be made available by the authors.

Author contributions:

Experimental concept and design: UR and JA (IOCAG)

Execution of the experiment: All authors.

Data analysis: DGS, AGG, MGD, JMSC

Manuscript preparation: DGS.

Manuscript revisions: All authors.

Financial Support

This study was funded by the OceanNETS project ("Ocean-based Negative Emissions Technologies – analysing the feasibility, risks and co-benefits of ocean-based negative emission technologies for stabilizing the climate", EU Horizon 2020 Research and Innovation Programme Grant Agreement No.: 869357), and the Helmholtz European Partnering project Ocean-CDR ("Ocean-based carbon dioxide removal strategies", Project No.: PIE-0021) with additional support from the

AQUACOSM-plus project (EU H2020-INFRAIA Project No.: 871081, "AQUACOSM-plus: Network of Leading European AQUAtic MesoCOSM Facilities Connecting Rivers, Lakes, Estuaries and Oceans in Europe and beyond").

Competing interests

The authors declare that they do not have any competing interests and that all of them agree to the content of this manuscript.


Acknowledgements



We would like to thank the Oceanic Platform of the Canary Islands (PLOCAN) and its staff for the use of their facilities and for their help with the logistics and organisation of this experiment. We are also grateful to Andrea Ludwig, and Jana Meyer (KOSMOS Logistics and Coordination), to Jan Hennke, Michael Krudewig, and Anton Theileis (KOSMOS technical staff),

to Michael Sswat, Carsten Spisla, Daniel Brüggemann, Silvan Goldenberg, Joaquin Ortiz, Nicolás Sánchez, and Philipp Suessle (KOSMOS Scientific Diving and Maintenance Team).