# Peer review of "Ocean alkalinity enhancement using sodium carbonate salts does not lead to measurable changes in Fe dynamics in a mesocosm experiment"

_EGUsphere, 2023_

## Author Comment (AC1)

**Reviewer 1**

**RC1: 'Comment on egusphere-2023-2868', Anonymous Referee #1, 18 Jan 2024  reply**

**This study by González-Santana et al. attempts to assess the impact that the addition of sodium carbonate and sodium bicarbonate minerals to seawater mesocosms has on iron dynamics. The study presents new measurements of the different iron species over the course of the addition of alkalinity into the mesocosms. Unfortunately, the mesocosms do not look to have been setup to study this as a specific aim, which leads to several shortcomings in the research. As the authors point out the mesocosms were not carried out following trace-metal requirements, which will have substantially impacted their data. In general, the data itself looks very noisy, with no trends that would be statistically significant within the given dataset. Given the limitations in the field data, the authors could have conducted laboratory experiments or even simple biochemical modelling to compare with the data to see whether we would even expect to see any changes in the data given the limited impact that these alkalinity additions have on seawater chemistry.**

**The other major issue that I have with the study is that one of the conclusions is that the iron size fractions and phytoplankton are independent. This is clearly the case as this is not an iron-limited environment. If this study had been conducted in an iron-limited environment, then the results might have been very different.**

**While I am not opposed to publishing a null result (which I believe these to be), the authors could have compared the results to laboratory data or a biochemical model which would greatly strengthen this work.**

Dear reviewer,

We completely agree with the fact that these experiments were not carried out following trace metal procedures in iron limited environments. Those conditions would probably have led to different results where iron additions in iron limited conditions could affect primary productivity and the evolution of the mesocosms. However, our goal was to observe what happened with the iron in coastal mesocosms experiments in an environment where coastal marine Carbon Dioxide Removal techniques have been proposed.

We did not look into the use of biochemical models to explain the possible results as surface waters present significant Fe(II) concentrations which we currently cannot account with our theoretical equations and Fe(II) oxidation kinetics constant equations. We think that the use of a biogeochemical model could lead to a second article where the model is the center of the manuscript, and it tries to explain the results within the noise observed in our data.

**11: OAE should be Ocean Alkalinity Enhancement. (Not alkalinization; alkalinization enhancement does not make sense).**

Done

**15: replace levels with additions.**

Done

**17: add a capital delta symbol before 2400.**

Done

**20: delete "could" and "because of".**

Done

**21: add "due to" before OAE.**

Done

**32: OAE should be Ocean Alkalinity Enhancement. (Not alkalinization; alkalinization enhancement does not make sense). Replace "both produce" with "lead to".**

Done

**37: replace "might have" with "has" and "consequences in" with "consequences for".**

Done

**38: replace "heightening" with "increasing".**

Done

**39: The hydrogen ion concentration is the pH (which you've already said increases!).**

Removed

**40: replace "hoisted" with "increased".**

Done

**45: Where are the proposed mineral dissolution products coming from? Are you suggesting the sediments, or biogenic materials in the water column, or something else entirely? Add "e.g." before the list of potential minerals as more than these three are used.**

We have added e.g. and added the references: Renforth and Henderson, 2017; Hartmann et al., 2013; Bach et al., 2019.

**46: "necessary" not "necessitated".**

Done

**47: "technique under" not "technique at".**

Done

**49: Replace "factor" with "variable of interest".**

Done

**64: delete "time".**

Done

**65: delete "the" before photooxidation and decreasing.**

Done

**66: delete "in the".**

Done

**67: rephrase to "All of these factors can induce…"**

Done

**71: "colloidal size fraction…"**

Done

**75: replace "a significant higher" with "significantly higher".**

Done

**78: replace "by OAE" with "for OAE".**

Done

**82: The mass of Ca or Mg required to impact the amount of Mg and Ca in seawater makes this a negligible impact (~2.5 mM DIC, ~10 mM Ca, ~50 mM Mg). No one is proposing the scale of OAE that would impact Ca and Mg concentrations enough to impact iron.**

**87: Again, think about the mass balance and consider how much alkalinity enhancement you would have to do before impacting the concentrations of Ca and Mg! Even in these experiments with a doubling of the alkalinity (which is extreme!) this would only impact Ca by ~25% and Mg by ~5%.**

Answer to L82 and L87.

We agree that the added Ca and Mg would not drastically change their relative concentration since they are major ions in seawater. However, in Santana-Casiano et al., (2010) they showed that the combined effect of calcium/magnesium in seawater with changes in pH could have an effect on Fe-organic compound interactions and speciation. Therefore, if there is an initial salt which should be considered first in mesocosms experiments it should be a Na salt due to its lower impact in the iron cycle. However, more research should be performed considering how these salts affect iron interactions before performing large scale experiments.

**89: "preceptive" is not what you mean here?**

We have changed the word to "recommended".

**90: Stick to what you have done in this paragraph and simplify.**

We have reworded the whole paragraph and simplied it to: "The aim of this research was to study the evolution of inorganic and organic Fe speciation in seawater under OAE scenarios. We considered if the added material acted as a significant iron source, and its effects on both the iron size fractionation and Fe-binding ligand evolution."

**92: delete "previously".**

Done

**93: replace "used alkalinization methodology" with "added material" and "which" with "what".**

Done

**111: add capital delta symbol before 2400.**

Done

**118: replace "Subsequently, 0.2 um filtered samples followed on" with "Aliquots were filtered (0.2 uM SartobranTM PES) for sFe, dFe and LFe.".**

Done

**120: delete the bracket after sampling and add "for" after "dark".**

Done

**121: replace "to" with "at".**

Done

**141: Define "TAC" here.**

We have changed as stated in comment L150.

**142: Do you mean +0 Fe additions?**

We have modified the sentence to: "starting with two no Fe additions followed by more than 10 titration points (Gledhill and Buck, 2012; Garnier et al., 2004)."

**150: Use this definition of TAC above, not here.**

Done

**163: Add a space before units (do this throughout the paper, it is currently inconsistent).**

We have gone through the manuscript and found two more cases.

**194: Superscript "-1".**

Done and checked other $^{-1}$ superscripts.

**196: I don't think this is significant given the noise in the rest of the data.**

**218: I don't think the data conclusively shows that the cFe concentration was directly impacted by the addition of the sodium carbonates given the noise in the dataset.**

Answer to comments in L196 and L218

Given the noise and average concentrations through the experiment, it does not initially seem significant. However, all the analysed samples converge towards a similar concentration, and it is what we wanted to indicate.

**211: % not ‰.**

Following GEOTRACES cookbook, we acidify our samples to 2‰ v/v HCl (pH ~2.0). We have modified the sentence to specify this.

**213: This sentence is unfinished.**

This was a sentence that was left here when we moved it to a different section of the manuscript in our pre-submission version. We have eliminated this sentence.

**216: replace "due to its" with "as it is".**

Done

**217: replace "uptake" with "to uptake".**

Done

**236: This is unsurprising given that these experiments were conducted in an environment where iron is not limiting!**

**255: replace "previous to" with "before".**

Done

**258: replace "concerning D1500" with "in the D1500 mesocosm".**

Done

**263: replace "computed" with "calculated to be".**

Done

**269: The values in this paragraph don't seem that similar. Be clearer and specifically compare values.**

We have modififed the paragraph and added "The pumping of more coastal water to fill the mesocosms could explain the observed increase in concentrations."

**273: superscript "-1".**

Done

**277: "being the weaker ligands the most abundant" does not make sense.**

We have modified the sentence to "being weaker ligands ($L_2$) more abundant than stronger ligands ($L_1$)"

**282: remove "a" before "biological".**

Done

**288: replace "from" with "of".**

Done

**300: rephrase to "biological parameters. This suggest the alkalinity addition in the mesocosm experiments did not…".**

Done

**310: The sentence starting with "Where the…" needs to be rephrased.**

We have combined and modified this sentence with the previous one: "This could have been a result of the manipulation required to fill in mesocosms in the open environment, where the iron sources such as sediment resuspension, aerial input and water runoffs are combined with our mesocosm manipulation."

**312: "concentration" not "concentrations".**

Done

**313: I still don't think this has been shown.**

**We have modified the sentence to: "**Nevertheless, the addition of sodium carbonate salts ($NaHCO_3$ and $Na_2CO_3$) coincided with a decreases in the dFe concentration just after the addition. **"**

**315: "change" instead of "present changes".**

Done

**316: "were added" not "addition" and "may" instead of "will shortly".**

We have changed the sentence to "The results suggest that the added sodium carbonate salts may…"

**318: This paragraph is discussion not conclusion as it is not related to what you did in these experiments.**

We agree with the reviewer and have moved the paragraph to the end of the discussion.

**327: delete "obtained".**

Done

**328: delete "i.e., did not alter the Fe cycle," as it is just a repetition.**

Done

**329: replace "was" with "were".**

Done

**330: delete "extensions" and reword the following two sentences.**

We deleted the word extension and modified the last two sentences to "However, it's important to consider that while employing this Oceanic Iron Fertilization (OAE) technique in coastal waters might not significantly impact the iron cycle, outcomes could differ under varying conditions. For instance, if the experiment were conducted in oligotrophic open ocean waters, the presence of a bloom exceeding those observed in these experiments could lead to different results. Conversely, an excess of iron might inhibit phytoplankton growth due to potential toxic effects. Therefore, further research into the effects of Carbon Dioxide Removal (CDR) techniques on trace metals is essential to mitigate unintended consequences on the ecosystem."

**Figures 2 and 3: Both figures need error bars.**

We have added the error bars to the figures, see some examples below. We do not see an improvement in the information presented. In most cases the errors are within the symbol size. The increase in the number of lines is making it a bit excessive. We are not against adding error bars. We would like you to confirm that the addition of the error bars provides more pros than cons.

---

## Author Comment (AC2)

**Reviewer 2**

RC2: 'Comment on egusphere-2023-2868', Anonymous Referee #2, 22 Jan 2024  reply

The manuscript entitled "Ocean alkalinity enhancement using sodium carbonate salts does not impact Fe dynamics in a mesocosm experiment" by Gonzalez-Santana et al., is an interesting manuscript in which a mesocosm experiment to study the effect of Ocean Alkalinity Enhancement over the iron fractionation and other physicochemical and biological variables is evaluated. Although the authors indicate that 1) some contamination problems could have happened because the experiment was not conducted under stringten trace trace metal conditions according to the GEOTRACES protocol and 2) the main conclusion is that "The iron size fractionation, concentration and iron-binding ligands data obtained supports the fact that the addition of sodium salts in this mesocosm experiment did not lead to significant changes in the iron cycle, i.e., did not alter the Fe cycle, therefore phytoplankton was not affected by changes in this essential element", in my humble opinión the present work deserves to be published after some minor changes. The purpose of the study (iron cycle under environmental chnaging conditions) is of great interest for the scientific community.

We would like to thank the reviewer for their inputs on our manuscript. We have accepted all minor comments requiring small modifications in the text and added a "Done" in each comment so as to confirm the modification. Answers to comments that required longer modifications are explained below each comment.

**Minor comments:**

**Page 1, line 15. "consisting on the controlled variation of total…"**

Done, we have modified the sentence.

**Page 1, line 19. The differences between TdFe and dFe should be explained.**

We have added a description of each. The new sentences is "Iron (Fe) speciation was monitored during this experiment to analyse whether total dissolved iron (TdFe, unfiltered samples), dissolved iron (dFe, filtered through a 0.2 µm pore size filter), soluble iron (sFe, filtered through a 0.02 µm pore size filter)…"

**Page 1, lines 22 and lines 28. In my humble opinión, these messages are contradictory. "There were variations in Fe size fractionation…" and …"this mesocosm experiment did not modify iron dynamics…"**

After the addition of the carbonate salts, there were changes in the Fe size fractionation (Figures 2 b and c). However, conditions returned to background levels within the next sampling period.  We have modified the text to clarify.

**Page 6, line 120. The symbol ")" after sampling should be deleted.**

Done

**Page 9, line 195. The symbol "." After mesocosm should be deleted.**

Done

**Page 9, line 213. Some information is missing after "Their variability…."**

We have eliminated this sentence. It was left from a previous version of the manuscript.

**Page 10, line 230. Authors talk about sediment resuspension. Is the mesocosm open in the bottom to consider this possibility? Please explain.**

The idea behind the sentence was considering the initial water collection that would contain particulate iron within the samples. There could potentially be a small iron source due to opening the mesocosms during individual sampling. We have expanded the sentence so as to consider these factors "…which would increase initial Fe concentrations used in the mesocosms compared to open ocean locations."

**Page 10, line 239. This argument would be enriched by including the following study:**

**Cabanes, D.J.E., Norman, L., Santos-Echeandía, J., … Laglera, L.M., Hassler, C.S., 2017. First evaluation of the role of salp fecal pellets on iron biogeochemistry. Frontiers in Marine Science, 2017, 3(JAN), 289.**

We have added the recommended reference which fits nicely with the ideas in the paragraph.

**Page 12, line284. This statement is only true for treatment Δ1500.**

We agree, we have modified the sentence to make this clear. The great increase was seen in this treatment for both parameters.

**Page 12, line 301. Please change "….and  biological" by "…or biological"**

Done.

**Page 13, line 306. This statement is only true for treatment Δ1500.**

We have modified the statement to make it clear that this was observed in the Δ1500 treatment: "…particularly during the emergence of nanoeukaryotes blooms and peaks of Synechococcus in the Δ1500 treatment…"

**Page 13, lines 316 and 317. Could this behaviour be associated to the buffering capacity of seawater?**

Initially we thought the same. However, this behavior is observed in all mesocosms without having a correlation with the increase in alkalinity (and salt addition), therefore there should be some other factor influencing the decrease in cFe which is aggregated towards the pFe fraction.

We have added : "Nevertheless, the observable decrease is not proportional to the increase in alkalinity. Where other factors such as aggregation due to increases in particles or added mineral salts produce a short term cFe decrease."

**Page 13, line 319. What does CDR mean? Please explain.**

It was explained in line 34 but not used again. We understand how it is very far from the first mention, so we have added the definition in this line.

**Page 13, line 321. Please insert a space before "Also….".**

Done

**Page 13, lines 320-321. comparison between the study carried put by Santana-Casiano et al., 2010 and the present study in which different salts that make the water more alkaline are added would be of great interest.**

In the Santana-Casiano et al. (2010), the researchers investigated the effect of the major seawater ions on the Fe(II) regeneration. They observed that calcium and magnesium competed with Fe for the available organic compounds. On the other hand, sodium does not strongly interact with organic compounds.

We are planning future research studies considering olivine and calcium carbonate minerals where the studies following Santana-Casiano et al. (2010) will be required as explained by the reviewer. In the text we make a reference to this article so as to alert future mesocosm or natural experiments where other salts are added.

**Citation: https://doi.org/10.5194/egusphere-2023-2868-RC2**

---

## Author Response (AR2)

Dear Editor,

We would like to thank you for your work during the whole process of reviewing our article. We have performed all the recommendations you indicated in your response.

Thanks

David et al.

**After receiving two largely positive reviews, the authors addressed the reviewers' concerns and suggestions in a comprehensive way. The paper now appears ready to move into a final editing stage to respond to the editor's remaining suggestions.**

**I have only one major suggestion, which is to bring the title more in line with the analysis. Something along the lines of**

**"Ocean alkalinity enhancement using sodium carbonate salts does not lead to measurable changes in Fe dynamics in a mesocosm experiment,"**

We have changed the title to the recommended one.

**Because the system has so much variability, and the possibility of contamination from various sources, I think it the narrower title is more in line with the results and conclusions from the work.**

**And I make the following minor suggestions. Several of these are stylistic and aim to help improve the readability for greatest impact:**

**1) I am confused by this statement (Line 40), as the increase in pH is erased by the CDR, which is the ultimate goal of an OAE intervention. So I don't think successful CDR can be said to reverse ocean acidification.**

**> Moreover, the increased pH and carbonate ion concentration reverses ocean acidification due to the uptake of anthropogenic CO2 from the atmosphere (Lenton et al., 2018).**

**You could simply eliminate the sentence, as it's not central to the paper, or better explain.**

Done

**Line 33 - eliminate the clause "is a negative emission technology (NET) that," as you only use the acronym NET once more in the paper (line 49, where it could easily be replaced with mCDR): Artificial Ocean Alkalinity Enhancement (OAE) can lead to the atmospheric carbon dioxide removal (CDR) by increasing the ocean carbon uptake and reversing ocean acidification (OA), favoring a higher ocean pH buffering capacity (Kheshgi, 1995; Renforth and Henderson, 2017; Lenton et al., 2018).**

Done

**Line 89 - OAE is misspelled AOE. Also replace the comma with the word "or" here: Mg2SiO4, CaO**

Done. We rechecked the whole document and found out one more misspelled OAE.

**Line 259 = add clean to "trace metal clean"**

Done